# Screening and Activity Analysis of α-Glucosidase Inhibitory Peptides Derived from Coix Seed Prolamins Using Bioinformatics and Molecular Docking

**DOI:** 10.3390/foods12213970

**Published:** 2023-10-30

**Authors:** Zhiming Li, Shu Zhang, Weihong Meng, Jiayu Zhang, Dongjie Zhang

**Affiliations:** 1College of Food, Heilongjiang Bayi Agricultural University, Xinfeng Lu 5, Daqing 163319, China; lizhiming1998@126.com (Z.L.); zshu996@163.com (S.Z.); mengweihong0423@126.com (W.M.); zjy_1997@outlook.com (J.Z.); 2National Coarse Cereals Engineering Research Center, Daqing 163319, China; 3Key Laboratory of Agro-Products Processing and Quality Safety of Heilongjiang Province, Daqing 163319, China

**Keywords:** hypoglycemic peptide, coix seed prolamins, α-glucosidase, structure-activity relationship, stability of activity

## Abstract

Hydrolysates of coix seed prolamins (CHPs) have an excellent hypoglycemic effect and can effectively inhibit α-glucosidase, which is the therapeutic target enzyme for type 2 diabetes mellitus. However, its hypoglycemic components and molecular mechanisms remain unclear, and its stability in food processing needs to be explored. In this study, four potential α-glucosidase inhibitory peptides (LFPSNPLA, FPCNPLV, HLPFNPQ, LLPFYPN) were identified and screened from CHPs using LC-MS/MS and virtual screening techniques. The results of molecular docking showed that the four peptides mainly inhibited α-glucosidase activity through hydrogen bonding and hydrophobic interactions, with Pro and Leu in the peptides playing important roles. In addition, CHPs can maintain good activity under high temperatures (40~100 °C) and weakly acidic or weakly alkaline conditions (pH 6.0~8.0). The addition of glucose (at 100 °C) and NaCl increased the inhibitory activity of α-glucosidase in CHPs. The addition of metal ions significantly decreased the inhibitory activity of α-glucosidase by CHPs, and their effects varied in magnitude with Cu^2+^ having the largest effect followed by Zn^2+^, Fe^3+^, K^+^, Mg^2+^, and Ca^2+^. These results further highlight the potential of CHPs as a foodborne hypoglycemic ingredient, providing a theoretical basis for the application of CHPs in the healthy food industry.

## 1. Introduction

Diabetes is a chronic metabolic disease with a high incidence worldwide, manifested as abnormal fluctuations in blood sugar, accompanied by limited insulin secretion and insufficient sensitivity. The number of patients is rising year by year [1]. Type II diabetes mellitus (T2DM), as the main type of the disease, accounts for 90–95% of the incidence of diabetes. Its typical characteristic is persistent hyperglycemia or postprandial hyperglycemia. Finding food-derived active ingredients to regulate blood glucose level represents an important strategy to manage T2DM [1]. α-glucosidase (EC 3.2.1.20) is a digestive enzyme located at the brush border of the epithelial cells of the small intestine. Its main function is to break down or transform dietary carbohydrates, release free glucose, and cause the glycemic index to rise after the glucose is absorbed and enters the circulation system [2]. The use of inhibitors to control α-glucosidase activity can not only effectively maintain blood glucose homeostasis, but also relieve the working pressure of islet tissue to a certain extent, which is an efficient and direct T2DM management strategy [3]. α-glucosidase inhibitors such as acarbose, voglibose, and other synthetic agents have a good curative effect, but they can easily cause abdominal distension, diarrhea, stomach colic, and other intestinal discomfort [4]. Food-derived α-glucosidase inhibitory peptides have become a hot research topic due to their natural properties, high safety, strong biocompatibility, and long-term consumption. Therefore, it is necessary to select a suitable food-derived protein to prepare high-activity α-glucosidase inhibitory peptide.

Coix seed (*Coix larchryma-Jobi* L. *var. ma-yuan Stapf*) is a kind of gramineous plant that can be used as both food and medicine. It has the pharmacological effect of strengthening the spleen, suppressing dampness, and improving T2DM disease [5]. Studies have shown that coix seed proteins have the effect of regulating blood sugar. Coix seed proteins can improve T2DM in mice by repairing islet cells, regulating in the IKK/NF-B signaling pathway, or mediating lipid metabolism and oxidative stress in vivo [6,7]. At present, some peptides with physiological activity are often encoded in protein sequences and are further released by metabolic enzymes after ingestion. Some specific configurations of bioactive peptides have a variety of physiological effects, such as maintaining blood glucose homeostasis, regulating blood pressure balance, anti-inflammatory effects, and other important health functions [8]. The bioactive peptides prepared by coix seed prolamins (the main components of coix seed protein, accounting for 44.74%) have the effects of lowering blood pressure, protecting the liver, and lowering blood glucose [9]. Bioactive peptides derived from coix seed prolamins include angiotensin-converting-enzyme inhibitors [10] and hepatoprotective peptide [11]. In addition, coix seed prolamins are rich in Pro, Leu, Ala, and other amino acids. Studies have shown that the inhibitory activity of α-glucosidase is enhanced when the proportion of these amino acids in the active peptide is high [12,13]. The sequence of coix seed prolamins also shows that some peptide fragments have the structural characteristics of α-glucosidase inhibitory peptides [12,13], which is a potential source of hypoglycemic peptides and needs to be developed and studied urgently. To date, the specific configuration and mode of action of α-glucosidase inhibitory peptides derived from coix seed prolamins have not been studied in depth. In addition, when glycemic peptides are applied to foods, their hypoglycemic activity may be affected by processes such as heat treatment, pH modification, and other components of the food substrate such as sugars and minerals. Therefore, it is necessary to investigate the activity stability of the hydrolysate of coix seed prolamins (CHPs) before it is used in food processing.

Therefore, in this research, we explored the influence of six proteases on the α-glucosidase inhibitory activity of CHPs and selected the enzyme with the strongest inhibitory activity to prepare α-glucosidase inhibitory peptides. The active peptides were identified by LC-MS/MS, and the molecular mechanism of inhibition of α-glucosidase by peptides was explained by molecular docking technology. Finally, the activity stability of CHPs against pH, thermal, and common food substrates such as sugar, salt, and metal ions was evaluated.

## 2. Materials and Methods

### 2.1. Materials and Reagents

Coix seeds were purchased from Ganzhou Kangrui Agricultural Products Co., Ltd. (Ganzhou, China). The dithiothreitol (DTT) and other analytical reagents were all purchased from Sigma-Aldrich Chemical Co. (St. Louis, MO, USA). The α-glucosidase and *p*-nitrophenyl-d-glucopyranoside (pNPG) were purchased from Shanghai Yuanye Biotechnology Co., Ltd. (Shanghai, China). The citric acid, sodium benzoate, potassium sorbate, lemon yellow, carmine, glucose, sucrose, and xylitol (food grade) were purchased from Henan Wanbang Chemical Technology Co., Ltd. (Zhengzhou, China). The alcalase (CAS: 9014-01-1, 200 U/mg), flavourzym (CAS: 9001-92-7, 30 U/mg), papain (CAS: 9001-73-4, 100 U/mg), neutral (CAS: 9068-59-1, 200 U/mg), trypsin (CAS: 9002-07-7, 250 U/mg), and compound protease (CAS: 9014-01-1, 100 U/mg) were purchased form the Beijing Solarbio Technology Co., Ltd. (Beijing, China).

### 2.2. Preparation of Coix Seed Prolamins

The method of Liu et al. [14] was referred to and modified slightly. Firstly, 80% ethanol solution, 1 mg/mL DTT solution, and 3 mol/L sodium acetate solution were prepared according to a volume ratio of 100:1:1. The ratio of material to liquid was 1:6. The mixture was well mixed and placed in the ultrasonic cleaner (SB25-12DTD Ultrasonic Cleaner; Ningbo Xinzhi Biotechnology Co., Ltd., Ningbo, China) for pre-treatment (250 W, 40 kHz, 40 °C, 30 min). The supernatant was centrifuged under centrifugation conditions of 3000× *g* for 20 min. Finally, equal volume of deionized water was added to the supernatant and precipitated. The precipitate was freeze-dried to obtain the coix seed prolamins.

### 2.3. Preparation of Coix Seed Prolamins Hydrolysate (CHPs)

In the present study, we selected six commercial proteases for the enzymatic hydrolysis of coix seed prolamins. Firstly, coix seed prolamins were suspended in water (1:25 *w*/*v*). The coix seed prolamins suspension was treated using a water bath at 90 °C for 30 min, and then an ultrasonic cleaning device was used for ultrasonic treatment at 300 W, 40 kHz, and 25 °C for 30 min. Then, equal amounts of proteases were added to the solution at 8000 U/g, respectively, time of hydrolysis 2 h, with pH and temperature optimal for each protease [15]: alcalase (pH 9.0, 50 °C), flavourzyme (pH 7.0, 53 °C), papain (pH 8.0, 60 °C), neutral (pH 7.0, 45 °C), trypsin (pH 7.5, 37 °C), and compound (pH 6.0, 45 °C). Protein hydrolysis was stopped by heating 100 °C for 10 min, and centrifuged at 3500× *g* for 20 min to obtain supernatant liquor.

### 2.4. Degree of Hydrolysis (DH) Assay

According to Feng et al. [16], *DH* was calculated using the method of pH-stat as follows:(1)$$DH=\frac{V\times N}{\alpha \eta m{ℎ}_{tot}}\times 100\%$$
(2)$$\alpha =\frac{10^{pH-pk}}{1+10^{pH-pk}}$$
(3)$$pk=7.8\times \frac{298-T}{298\times T}\times 2400$$
where *V* is the amount of NaOH consumed (mL), *N* is the concentration of the NaOH solution (mol/L), *η* is the total protein content (g/g) in coix seed prolamin samples, *m* is the mass of the coix seed prolamins sample (g), *h_tot_* is the number of millimoles of peptide bonds per unit mass of raw barley protein (8.3 mmol/g), and *T* is the test temperature.

### 2.5. α-Glucosidase Inhibitory Activity Assay

Determination of α-glucosidase inhibition was carried out using the pNPG approach, referring to the method provided by Huang et al. [17]. Four groups were set up, which were blank group, blank control group, sample group, and sample control group. The experiment was conducted on a 96-well plate. The absorbance values measured by each group after sample addition were defined A_c_, A_b_, A_s_, and A_n_, respectively. For details, see Table 1. Equation (4) was used to calculate the α-glucosidase inhibitory activity:(4)$$\alpha -glucosidase\;inhibitory\;activity=\left(1-\frac{A_{s}-A_{n}}{A_{c}-A_{b}}\right)\times 100\%$$

### 2.6. Identification of the Peptides (LC-MS/MS)

The CHPs were first desalted using the C18 StageTip and then vacuum dried. All chromatographic analyses were performed on the nanoliter flow rate Easy nLC 1200 chromatography system (Thermo Scientific, Waltham, MA, USA). The mobile phase A was 0.1% formic acid solution. The mobile phase B was 0.1% formic acid acetonitrile solution. The peptides were separated from 2% to 100% mobile phase B in 60 min at a flow rate of 300 nl/min. Subsequently, the isolated peptides were analyzed using mass spectrometry with Q-Exactive Plus mass spectrometer (Thermo Scientific, Waltham, MA, USA). The conditions of mass spectrum are as follows: (1) the detection method is positive ion; (2) primary mass spectrometry: scanning range (*m*/*z*) is 350–1800; resolution: 60,000; automatic gain control (AGC) target: 3 × 10^6^; maximum injection time: 50 ms; (3) secondary mass spectrometry: resolution: 15,000; automatic gain control (AGC) target: 1 × 10^5^; maximum injection time: 50 ms; isolation window: 1.6 *m*/*z*; normalized collision energy: 28. MaxQuant (version 2.1.2.0) software combined with UniProt database was used for mass spectrometry retrieval.

### 2.7. Bioinformatics Analysis of Bioactive Peptides (BPs) Screening from CHPs

The bioinformatics approaches allow for the screening and analysis of potential bioactive peptides [18]. In this study, the probability of the peptide with bioactivity was predicted by PeptideRanker, available at http://distilldeep.ucd.ie/PeptideRanker/ accessed on 15 July 2023, where the higher the score, the higher the probabilities of the peptide being bioactive. Peptide novelty check using databases BIOPEP, PeptideDB, and EROP-Moscow was conducted. Moreover, the isoelectric point (pI), net charge (pH 7.0) and hydrophobicity of the peptides were analyzed in PepDraw web-servers, available at http://pepdraw.com accessed on 15 July 2023. The toxicity of peptides was predicted via the ToxinPred website, available at https://webs.iiitd.edu.in/raghava/toxinpred accessed on 15 July 2023.

### 2.8. Molecular Docking

The α-glycosidase crystal structure (PDB code: 2 QMJ) was the initial structure model of molecular docking. The crystal structure was obtained from the Protein Data (https://www.rcsb.org/structure/2QMJ, accessed on 20 July 2023) [19]. Before molecular docking, Pymol2.3.0 was used to remove the crystal water and original ligand of α-glucosidase, and AutoDocktools (v1.5.6) was used for hydrogenation, charge calculation, charge allocation, and atom type designation. The polypeptide structure mapped by ChemBioDraw Ultra 14.0 was used as a ligand, the α-glucosidase binding site was predicted using POCASA 1.1, and the molecular docking step was performed using AutoDock Vina1.1.2. The related parameters of α-glucosidase (2QMJ) were set as follows: center_x = −48.1, center_y = 13.1, center_z = −29.5; search space: size_x: 60, size_y: 60, size_z: 60 (spacing of each cell is 0.375A), and the remaining parameters were default settings. Finally, PyMOL2.3.0 and Ligplot V 2.2.8 were used to analyze the interaction pattern of the docking results.

### 2.9. Thermal Effects and pH Effects on the α-Glucosidase Inhibitory Activities of CHPs

The thermal and pH stability profile was determined according to Wong et al. [20], with some minor modifications. CHPs were dissolved in deionized water and configured into 10 mg/mL solution. The CHPs solution was incubated in a water bath at temperatures of 25 °C, 40 °C, 55 °C, 70 °C, 85 °C, and 100 °C for 30 min. At the end of thermal testing, the α-glucosidase inhibition of CHPs was determined.

Likewise, the pH of the configured 10 mg/mL CPHs solution was adjusted to 2, 4, 6, 7, 8, 10, and 12, followed by incubation at 25 °C for 30 min. The pH of the solution was adjusted to 7 at the end of the incubation, and the α-glucosidase inhibition rate was determined.

### 2.10. Effects of Food Ingredients on the α-Glucosidase Inhibitory Activities of CHPs

Here, we mainly investigated the effects of dietary sugars, NaCl, and metal ions on the α-glucosidase inhibitory activity of CHPs. First, the CHPs solution with a concentration of 10 mg/mL was configured, and then different amounts of food ingredients were added to the CHPs solution. The stability of activity for different edible sugars was determined. Certain amounts of glucose, sucrose, and xylitol were added to the solution of CPHs and brought to a concentration of 2, 4, 6, 8, and 10 g/100 mL. The solutions were incubated at 25 °C for 2 h and at 100 °C for 20 min, respectively. The stability of activity for different edible sugars and NaCl was determined. Certain amounts of NaCl and KCl, CaCl_2_, MgCl_2_, ZnSO_4_, CuSO_4_, and Fe_2_SO_3_ were added to CHPs solution to maintain their final concentrations at 50, 100, 150, 200, and 250 µg/mL for 2 h at 25 °C. The inhibition rate of α-glucosidase was measured after incubation.

### 2.11. Data Analysis

Data are presented as mean ± standard errors (*n* = 3). The statistical analysis was performed using IBM SPSS Statistics version 22.0. Data were analyzed using the ANOVA test, and means of significant differences were separated using Duncan’s test at the 0.05 level of probability.

## 3. Results

### 3.1. The DH and α-Glucosidase Inhibitory Activity of CHPs Generated by Different Proteases

To evaluate the ability of six commercial proteases to release α-glucosidase inhibitory peptides from coix seed prolamins, we performed hydrolysis and determined the DH and α-glucosidase inhibitory rates. As shown in Figure 1, the DH and α-glucosidase inhibition rates of CHPs obtained via alcalase hydrolysis were 32.47% and 67.26%, respectively, both of which were the maximum values. Alcalase is an endoprotease with broad specificity, preferentially cleaving C-terminal hydrophobic amino acid residues such as Leu, Ile, Typ, Val, Phe, and Met [21]. Analogous studies demonstrated that alcalase had very broad specificity in peptide cleavage and produced high values of DH when chickpea protein isolate, wheat gluten, and mung bean were hydrolyzed, with maximum DH values of 14.67%, 16.77% and 23.51%, respectively [22,23,24]. In addition, it should be noted that a high degree of hydrolysis does not mean high activity. Papain and trypsin showed a low DH (6.58% and 3.80%, respectively), but the α-glucosidase inhibitory activity of the hydrolysate was high (62.90% and 61.48%, respectively) (Figure 1). This may be related to the recognition of specific sites on the substrate by these two proteases, as papain and trypsin mainly tend to cut the C- terminal of the peptide to Arg or Lys [25]. The insufficient number of cleavage sites limits their cleavage range, but the resulting peptides tend to have high α-glucosidase inhibitory activity. It has been reported that polypeptides with Arg in the C-terminal can effectively bind to amino acid residues of α-glucosidase, thereby inhibiting the catalytic ability of α-glucosidase to the substrate [19]. Therefore, this reminds us that the screening of proteases is necessary in the preparation of active peptides. It is necessary to combine the amino acid profile of food proteins and the mechanism of protease action to match, in order to obtain the target product efficiently and accurately. The DH and α-glucosidase inhibition rates of CHPs obtained from the Alcalase group were the highest. In addition, the higher the activity and degree of hydrolysis, the higher the probability of preparing a small peptide with high activity and easy absorption, so only this hydrolysate was studied in the later stage. To distinguish the individual hydrolysates, the hydrolysates of Alcalase, Flavourzym, Papain, Neutrase, Trypsin, and Compound were represented as CHPs-Alc, CHPs-Fla, CHPs-Pap, CHPs-Neu, CHPs-Try, and CHPs -Com, respectively.

### 3.2. Identification of Peptides from CHPs-Alc

The peptides from CHPs-Alc were identified through LC-MS/MS. A total of 353 peptides were identified from CHPs-Alc (please refer to Appendix A for specific information). The identified peptides were composed of 5~30 amino acids, and the content of peptides with a length of 6~12 accounted for 79.04% (Figure 2A). Peptides are mainly derived from α-coixin (305), β-coixin (19), and γ-coixin (19) (Figure 2B). Coix seed prolamins are mainly composed of α-coixin, β-coixin, and γ-coixin, among which α-coixin is the most abundant and elastic, which is easily hydrolyzed by protease to obtain a large number of active peptides [14]. The molecular weight of peptides in CHPs-Alc was concentrated between 500 and 1500 Da, and there were 141 peptides with molecular weight less than 1000 Da, accounting for 39.94% (Figure 2B and Appendix A). From the above data, the chain length of peptides with molecular weight less than 1000 Da is less than nine (Appendix A), and this characteristic of chain length has two positive effects on the α-glucosidase inhibitory activity of peptides. First, because the α-glucosidase inhibitor peptide exerts its action at the brush border of small intestinal epithelial cells [26], the transport of peptides with chain length longer than four on small intestinal epithelial cells is usually transcytosis and passive transcellular diffusion, with limited transport efficiency [27,28]. This is conducive to the accumulation of peptides at the brush edge of small intestinal epithelial cells, and a greater probability of contact with α-glucosidase plays an inhibitory function. Second, for α-glucosidase inhibitory peptides with specific chain lengths (3–7), the binding effect with the active site of α-glucosidase is better, and the binding energy is lower [29,30]. Next, we screened highly active α-glucosidase inhibitory peptides via bioinformatics.

### 3.3. Screening of α-Glucosidase Inhibitory Peptides from CHPs-Alc via Bioinformatics

In this study, Peptide Ranker Score was combined with affinity to screen active peptides. First, Peptide Ranker Score was used to score the identified peptide, and a peptide greater than 0.5 was biologically active [31]. Then, the bioactive peptide (Peptide Ranker Score > 0.5) underwent molecular docking with α-glucosidase in the human body, and the binding energy of the interaction between peptide and α-glucosidase was obtained; the binding energy was sorted from largest to smallest. The results are shown in Appendix A. In total, 160 peptides were screened as bioactive peptides, with molecular weights (MWs) ranging from 609.31625 to 2161.2048 Da, isoelectric point (pI) ranging from 3.02 to 11.18, and net charge of 0 or +1 at pH 7.0. The hydrophobicity was in a range of 3.83~15.52 kcal/mol, and the peptide toxicity prediction of ToxinPred showed that all the 160 peptides were non-toxic.

The lower the energy required for the peptide to bind to α-glucosidase, the easier it is to bind to α-glucosidase [19]. Of the 160 biologically active peptides, 10 peptides (LFPSNPLA > FPCNPLV > HLPFNPQ > LLPFYPN > FLSPF > FPCNPLVA > LPPFLPS> QLFPSNPLA > QQHLPFNPQ > SWQQPIVGR) had a small binding energy with α-glucosidase (less than or equal to −8.4 kcal/mol) (Table 2), which was a potential α-glucosidase inhibitory peptide. Their secondary mass spectra and structures are shown in Appendix A. As shown in Table 2, except for QQHLPFNPQ and SWQQPIVGR, the molecular weights of other peptides were less than 1000 Da and belonged to small molecular peptides. The isoelectric point (pI) of the ten peptides varied from 5.21 to 10.85, and the net charge was mainly 0 (except SWQQPIVGR) (Table 2). The hydrophobicity was excellent and varied between 3.83 and 10.71 kcal·mol^−1^ (Table 2). According to the structure–activity relationship model, the activity of the α-glucosidase inhibitory peptide was closely related to amino acid composition and sequence, net charge, and hydrophobicity [32]. At present, the common peptide motif of the α-glucosidase inhibitory peptide has not been reported. After comparing the ten selected peptides with validated α-glucosidase inhibitory peptides such as soybean protein peptide (LLPLPVLK, SWLRL, and WLRL) [25], oat globulin peptide (LQAFEPLR and EFLLAGNNK) [33], and quinoa protein peptide (IQAEGGLT) [34], no primary structure homology was found between them. However, it has also been reported that the hydrophobic amino acid content has a significant positive effect on maintaining the high activity of α-glucosidase inhibitory peptides [35]. The contents of hydrophobic amino acids in the ten peptides ranged from 33.33% to 85.71%, except QQHLPFNPQ and SWQQPIVGR, and the contents of other hydrophobic amino acids were more than 50%. According to Ibrahim et al. [36] and Li et al. [37], Pro and Leu in peptides are key amino acid residues that exert α-glucosidase inhibition alone or synergically, and they are the molecular basis for the targeted screening and design of α-glucosidase inhibition peptides. Each of the ten peptides had at least one to three Pro or Leu, of which LPPFLPS had the highest “Pro + Leu” content, accounting for 71.43%. In summary, the ten peptides screened in this study had some common structural characteristics with the mainstream α-glucosidase inhibitory peptides.

### 3.4. The Interaction Mechanism between α-Glucosidase Inhibitory Peptides and α-Glucosidase

Further, we selected the top four active peptides with the highest binding energies (LFPSNPLA, FPCNPLV, HLPFNPQ, LLPFYPN) to focus on the molecular mechanism of α-glucosidase inhibition. LFPSNPLA, FPCNPLV, HLPFNPQ, and LLPFYPN bind to α-glucosidase mainly through two forces: hydrogen bonding and hydrophobic interaction. The number of action sites is between 19 and 24 (Figure 3 and Table 3). Studies have shown that the number of binding sites is positively correlated with the inhibitory effect of α-glucosidase inhibitory peptide, and stable peptide–enzyme complexes can be formed when the number is greater than eight [31]. It has been confirmed that hydrogen bonds play a key role in the stabilization of α-glucosidase action sites by active peptides [38]. Four peptides can participate in hydrogen bond interaction with amino acid residues of α-glucosidase. Each peptide binds to different active residues, but Arg730, Glu767, and Try733 are the most frequent ones (Figure 3 and Table 3). The hydrogen bond distance is between 2.71 and 3.49 A (Table 3), which is lower than the traditional hydrogen bond length of 3.5 A, and a larger number of strong hydrogen bonds is conducive to the peptide–enzyme binding effect [19]. In addition, LFPSNPLA, FPCNPLV, HLPFNPQ, and LLPFYPN did not bind to the active site of α-glucosidase described by Liu et al. [19]. However, the action mode of the four peptides was similar to that of camellia seed peptide (LLVLYYEY), and they all achieved binding with Arg730 through hydrogen bond interaction, which was consistent with the experimental results of Zhang et al. [39]. The results showed that the four peptides could inhibit the enzyme activity by interacting with the key amino acid residues (Arg730) of α-glucosidase. In addition, Pro and Leu in the peptide are key active groups, which play an important role in the formation of hydrogen bond forces. “Pro + Leu” in the four peptides interact with α-glucosidase via one to three hydrogen bonds, and the hydrogen bond length ranges from 2.89 to 3.49 (Figure 3). We can speculate that Pro and Leu are key amino acid residues that inhibit α-glucosidase activity.

### 3.5. Effect of Food Processing on α-Glucosidase Activity of CHPs-Alc

#### 3.5.1. Temperature and pH

When CHPs-Alc are incorporated into the food system as a functional component, it is necessary to consider the active stability of CHPs-Alc during food processing or storage. In conventional food processing, the identification of temperature and pH stability curves of active peptides helps to determine their suitability for heat treatment processing steps and food systems with specific acidity [40,41]. In general, after continuous heating at high temperatures, peptides with specific structures may undergo thermal aggregation, hydrolysis of peptide bonds, and reconstruction of disulfide bonds [42]. The original polypeptide chain structure was destroyed and the physiological activity of the polypeptide was affected [43]. As shown in Figure 4A, the α-glucosidase inhibitory activity of CHPs-Alc was the highest at 25 °C, and significantly decreased with the increase in temperature (*p* < 0.01). However, compared with 25 °C, it was found that the maintenance rate of α-glucosidase inhibitory activity was 85.95~93.78% in a temperature range of 40~100 °C. In addition, there was no significant change in the α-glucosidase inhibitory activity of CHPs-Alc at increasing temperature (70~100 °C) (*p* > 0.05). The above results showed that the α-glucosidase inhibitory activity of CHPs-Alc could still maintain a high level under high temperature heating conditions, and had high thermal stability. CHPs-Alc can better adapt to high temperature food processing or storage environments.

The change in pH can affect the ionization state of the polypeptide, and strong acids and bases can induce non-specific degradation of the polypeptide, resulting in changes in the structure and biological activity of the polypeptide [41,44]. As shown in Figure 4B, the α-glucosidase inhibition rate of CHPs-Alc was the highest in the pH 6~8 range, and there was no significant difference (*p* > 0.05). However, the α-glucosidase inhibition rate decreased significantly under acidic (pH < 6) and alkaline (pH > 8) conditions (*p* < 0.05). Compared with pH 7.0, the inhibitory activity of α-glucosidase decreased between 30.09% and 58.94% at pH < 6 and pH > 8. Under alkaline conditions, cysteine, serine, and threonine in polypeptides are easily destroyed, and racemization may occur, which ultimately leads to the reduction of α-glucosidase inhibitory activity [45]. Also, acidic conditions may influence the availability of hydroxyl groups on peptide molecules and improve the hydrogen donation ability of peptides [46]. This also reduces α-glucosidase inhibitory activity. Therefore, this suggests that when CHPs-Alc are used as a food substrate, a food system with a pH value of 6 to 8 should be considered first, and a strong acid and alkali environment should be avoided when processing.

#### 3.5.2. Edible Sugar

Glucose, sucrose, and xylitol are commonly used as food sweeteners in the food industry. The changes in the α-glucosidase inhibitory activity of CHPs-Alc after adding food sweeteners at different temperatures are shown in Figure 4C1 (25 °C) and Figure 4C2 (100 °C). At 25 °C, the addition of glucose, sucrose, and xylitol did not cause significant changes in the α-glucosidase inhibition rate of CHPs-Alc (*p* < 0.05) (Figure 4C1). This indicates that CHPs-Alc can coexist with these three edible sweeteners in the same system at room temperature. At 100 °C, the α-glucosidase inhibition rate of CHPs-Alc increased with the increase in glucose concentration. Compared with the control group, the α-glucosidase inhibition rate increased by 18.54% at 10 g/100 mL (Figure 4C2). In addition, the α-glucosidase inhibition rate of CHPs-Alc decreased gradually with the increase in sucrose and xylitol concentrations. At 10 g/100 mL, the loss of activity reached 25.83% and 23.89%, respectively (Figure 4C2). According to research reports, the carbonyl groups and polypeptides of reducing sugars may undergo condensation and Maillard reactions under heat treatment conditions [47]. The molecular weight distribution, surface hydrophobicity, and substance composition of CHPs-Alc are affected, and finally the α-glucosidase inhibitory activity of CHPs-Alc is changed [48]. Glucose is a reducing sugar, while sucrose and xylitol are non-reducing sugars. It is speculated that the increase in CHPs-Alc activity after the addition of glucose at 100 °C is caused by biochemical reactions such as the Maillard reaction, while the decrease in CHPs-Alc activity caused by the addition of sucrose and xylitol may be due to high temperature treatment. It also reminds us that when reducing sugars and peptides coexist in a food system and are processed, the effects of thermal processing operations on their interactions should be fully considered [40]. In summary, CHPs-Alc can co-exist with glucose, sucrose, and xylitol in a food system. Moreover, the α-glucosidase inhibitory activity of CHPs-Alc was enhanced via the addition of glucose under high temperature processing conditions (100 °C).

#### 3.5.3. NaCl

The activity stability of CHPs-Alc at different concentrations of NaCl is shown in Figure 4D. With the increase in the NaCl concentration, the α-glucosidase inhibition rate of CHPs-Alc increased gradually, from 0 μg/mL (58.01 ± 0.52)% to 250 μg/mL (71.54 ± 0.22)%, and the inhibition rate at 250 μg/mL was 1.23 times that of the blank group. The reason for this phenomenon may be that after the addition of NaCl, the ionized Na^+^ and Cl^−^ will destroy the charge balance of polypeptides in CHPs-Alc, resulting in a change in ionic strength in the solution system, thus enhancing their α-glucosidase inhibitory activity [49]. However, it has also been reported that too high a concentration of NaCl will cause damage to the amino acid side chain of polypeptides, affecting their structure and physiological activity [50]. Cys is capable of binding metal ions due to the thiol group, His can interact with metal ions due to imidazole ring [51]. Therefore, at high NaCl concentrations, changes in the configurations of Cys and His side chains lead to changes in α-glucosidase inhibitory activity. The final positive or negative effect depends on the composition and structure of the peptides in the protein hydrolysate. These results indicate that the addition of NaCl can ensure the high α-glucosidase inhibitory activity of CHPs-Alc in the food system with CHPs-Alc as one of the food substrates.

#### 3.5.4. Metal Ions

In the conventional food system, due to the addition of some salt or the use of metal containers, food ingredients inevitably come into contact with metal ions, such as K^+^, Ca^2+^, Mg^2+^, Zn^2+^, Cu^2+^, and Fe^3+^ [52]. When these metal ions and peptides coexist in a system, it will affect the ionization state of the peptide and change the interaction between the polypeptide–water molecules, and may also form a polypeptide-metal ion complex, which ultimately affects the activity of the peptide [53]. Moreover, different kinds of metal ions have different effects on the activity of the peptide, depending on the number of charges the metal ions carry [54]. The change in the α-glucosidase inhibition rate of CHPs-Alc after adding different metal ions is shown in Figure 4E. After adding K^+^, Ca^2+^, Mg^2+^, Zn^2+^, Cu^2+^, and Fe^3+^, the α-glucosidase inhibition rate of CHPs-Alc showed a decreasing trend. The inhibition rate of α-glucosidase of CHPs-Alc decreased with the increase in concentration, and the effect of different metal ions on α-glucosidase was different. Specifically, the degree of influence from large to small was Cu^2+^ > Zn^2+^ > Fe^3+^ > K^+^ > Mg^2+^ > Ca^2+^. Therefore, this also suggests that CHPs-Alc should avoid contact with Cu^2+^, Zn^2+^, and Fe^3+^ in food processing and storage to ensure the stability of the hypoglycemic activity.

## 4. Conclusions

In this study, four previously unreported α-glucosidase inhibitory peptides (LFPSNPLA, FPCNPLV, HLPFNPQ, and LLPFYPN) were identified and screened from CHPs (obtained by hydrolysis of Alcalase) via LC-MS/MS identification and virtual screening. The results of molecular docking showed that the peptide mainly binds to the active site of α-glucosidase through hydrogen bond and hydrophobic interaction force, in which Pro and Leu in the peptide played a key role in the formation of the hydrogen bond. In addition, the α-glucosidase inhibitory activity of CHPs remained high under heat treatment and pH 6.0~8.0 conditions. The addition of glucose (100 °C ambient temperature) and NaCl helped to enhance the α-glucosidase inhibitory activity of CHPs. However, the α-glucosidase inhibitory activity of CHPs decreased significantly after the addition of metal ions (from large to small Cu^2+^ > Zn^2+^ > Fe^3+^ > K^+^ > Mg^2+^ > Ca^2+^) (*p* < 0.05). These results indicate that CHPs can be used as a hypoglycemic component with α-glucosidase inhibitory activity, and they are expected to be used in health food in the future. In the future, we can focus on the hypoglycemic effect of CHPs in vivo, further clarify the hypoglycemic mechanism of CHPs through animal tests, and provide data and theoretical support for the development and application of CHPs.

## Figures and Tables

**Figure 1 foods-12-03970-f001:**
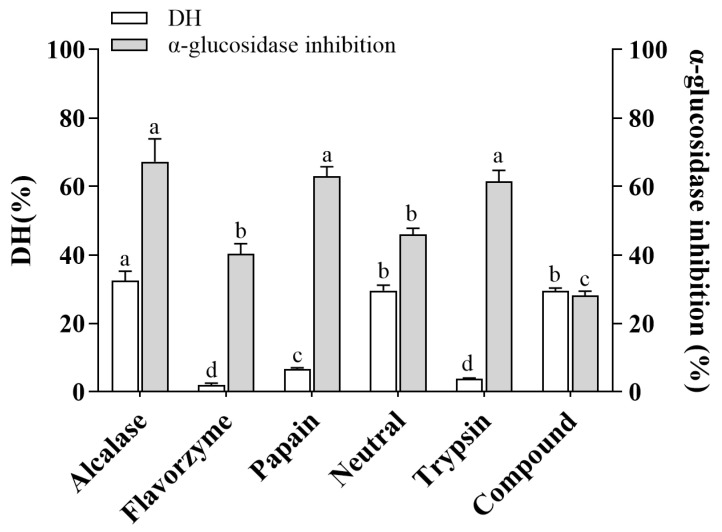
Effects of different proteases on the degree of hydrolysis and the inhibitory activity of α-glucosidase of the CHPs. Different letters Different letters (a–d) indicate significant difference between the mean values of different samples (*p* < 0.05).

**Figure 2 foods-12-03970-f002:**
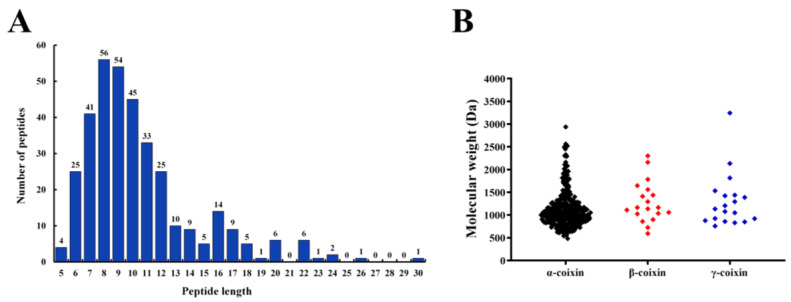
Peptides identified in the CHPs-Alc. (**A**) Peptides number of identified peptides from the parent proteins α-coixin, β-coixin, γ-coixin among all of the identified peptides in the CHPs-Alc; (**B**) molecular weight distribution of the identified peptides in the CHPs-Alc.

**Figure 3 foods-12-03970-f003:**
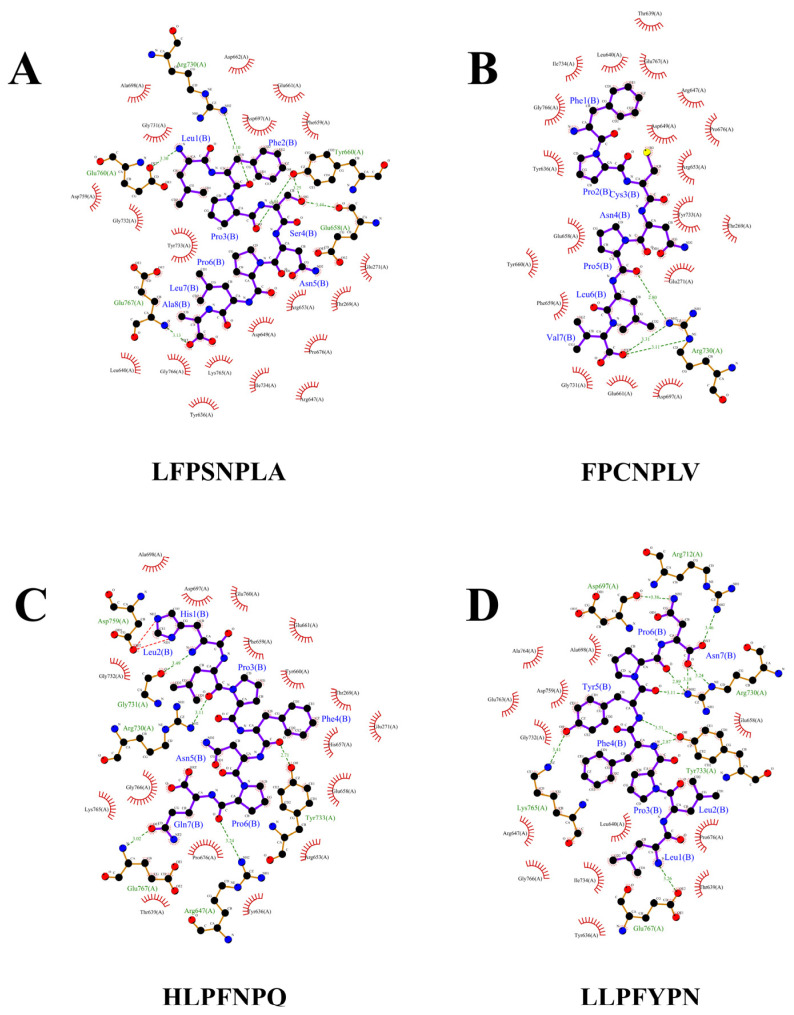
Molecular docking results of potential alpha-glucosidase inhibitory peptide with α-glucosidase (2QMJ). (**A**) LFPSNPLA; (**B**) FPCNPLV; (**C**) HLPFNPQ; (**D**) LLPFYPN.

**Figure 4 foods-12-03970-f004:**
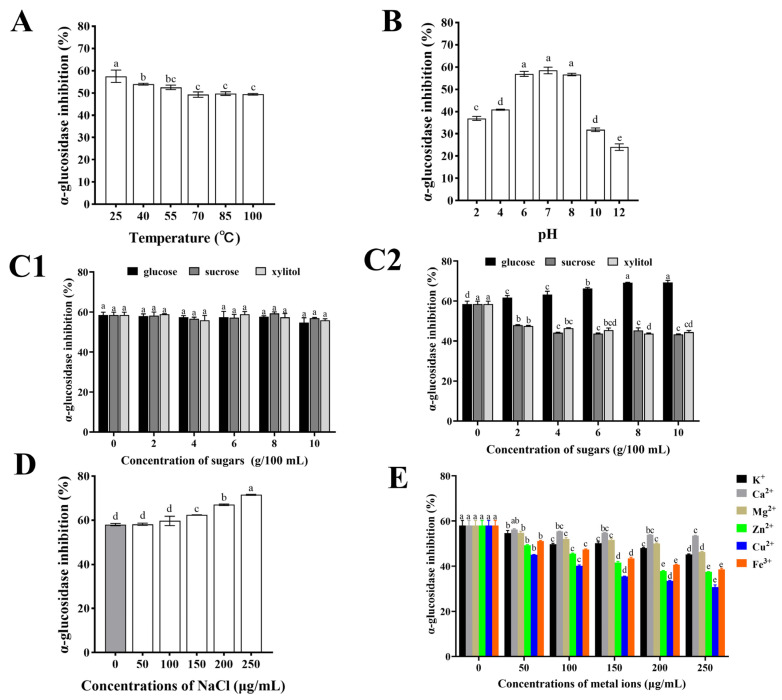
The stability of α-glucosidase inhibitory activity of CHPs-Alc under different food processing conditions: (**A**) Temperature, (**B**) pH, (**C1**) edible sugar was added at 25 °C, (**C2**) edible sugar was added at 100 °C, (**D**) NaCl, (**E**) metal ions. Different letters Different letters (a–e) indicate significant difference between the mean values of different samples (*p* < 0.05).

**Table 1 foods-12-03970-t001:** Procedure for determining α-glucosidase inhibitory activity.

Reagent	Blank Group/μL	Blank Control Group/μL	Sample Group/μL	Sample Control Group/μL
The solution of CHPs	/	/	50	50
α-glucosidase (0.1 U/mL)	50	/	50	/
Phosphate Buffered Saline (PBS) (0.1 mol/L, pH 6.8)	50	100	/	50
Mix well and incubate at 37 °C for 10 min				
pNPG (2.5 mmol/L)	50	50	50	50
Mix well and incubate at 37 °C for 30 min				
Na_2_CO_3_ (0.5 mol/L)	100	100	100	100
Absorbance was measured at 405 nm				

**Table 2 foods-12-03970-t002:** Potential α-glucosidase inhibitory peptides screening from CHPs-Alc.

NO.	Sequence	Length	MW/Da	Peptide Ranker Score	Affinity (kcal/mol)	pI	Net Charge at pH7.0	Hydrophobicity /(Kcal/mol)	Toxicy
1	LFPSNPLA	8	857.4647	0.648003	−8.9	5.6	0	5.78	-
2	FPCNPLV	7	845.41056	0.852117	−8.8	5.24	0	5.59	-
3	HLPFNPQ	7	851.42899	0.639342	−8.7	7.69	0	9.17	-
4	LLPFYPN	7	862.45889	0.81539	−8.7	5.39	0	4.11	-
5	FLSPF	5	609.31625	0.964433	−8.5	5.41	0	3.83	-
6	FPCNPLVA	8	916.44767	0.745656	−8.4	5.21	0	6.09	-
7	LPPFLPS	7	769.43743	0.850373	−8.4	5.45	0	4.57	-
8	QLFPSNPLA	9	985.52328	0.585702	−8.4	5.49	0	6.55	-
9	QQHLPFNPQ	9	1107.5461	0.574762	−8.4	7.59	0	10.71	-
10	SWQQPIVGR	9	1069.5669	0.703379	−8.4	10.85	+1	9.33	-

**Table 3 foods-12-03970-t003:** Estimates the binding energy and chemical interactions for the effective poses obtained by molecular docking of α-glucosidase inhibitory peptides.

NO.	Sequence	Binding Energy/kcal mol^−1^	Residues Formed Hydrogen Bonds with the Ligand	Number of Hydrogen Bonds	Distance of Hydrogen Bond	Number of Hydrophobic Amino Acid Residues
1	LFPSNPLA	−8.9	Glu658, Tyr660, Arg730, Glu760, Glu767	5	3.44Å, 3.25Å, 3.10Å, 3.30Å, 3.13Å	19
2	FPCNPLV	−8.8	Arg730	3	3.11Å, 3.31Å, 2.80Å	19
3	HLPFNPQ	−8.7	Arg647, Tyr733, Gly731, Arg730, Glu767	5	3.24Å, 2.71Å, 3.49Å, 3.11Å, 3.02Å	17
4	LLPFYPN	−8.7	Glu767, Tyr733, Arg730, Arg712, Asp697, Lys765	10	3.26Å, 2.87Å, 3.31Å, 3.11Å, 2.89Å, 3.18Å, 3.24Å, 3.46Å, 3.38Å, 3.41Å	13

## Data Availability

The data used to support the findings of this study can be made available by the corresponding author upon request.

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
