# Peer review of "Screening and Activity Analysis of α-Glucosidase Inhibitory Peptides Derived from Coix Seed Prolamins Using Bioinformatics and Molecular Docking"

_foods, 2023, doi:10.3390/foods12213970_

Round 1
Reviewer 1 Report
Comments and Suggestions for Authors
1) There are many expressions in the introduction that are incorrect as authors are trying to create their own. Some of those inaccurate expressions are provided in my assessment but the list is not exhaustive. Revision and rewriting are required.
2) The structure of sentences in general is poor as the same work or expression is often repeated within the same sentence or the next.
3) The rationale of testing the effects of sugars and mineral is not included in the manuscript. May not some information of that in the introduction.
4) How the previous work on the inhibition of α-glucosidase activity by hydrolysate of coix seed prolamins different from work included in the manuscript?
5) There of the hydrolysates had equivalent a-amylase inhibitory activities, only one was selected for further investigation without explanation.
6) pH stability: The hydrolyzed proteins were adjusted to 2 – 12 and then kept at room temperature (25 ℃) for 30 min and brought back to pH 7.0. Because they are assays at the same pH, one should not typically expect a difference. Lowering or increasing a pH and then bringing it back to the initial value should not typically affect the structures of peptides/proteins.
L34-36. Sentences should be reviewed to avoid repetition. E.g. the number of patients is rising … it is estimated that the number of patients or main type of diabetes … incidence of diabetes.
L37. The meaning is unclear: Exogenous regulators to regulate and intervene.
L49. I don’t understand the sentence: The deep excavation of their high-quality food sources is also key work in this field
L54. Delete coix seed before proteins and “the disease of”.
L55. Intervening doesn’t seem to be the right expression.
L56. Delete “which has great research value”.
L57. The main media expression is a wrong term to indicate dans peptides derived from proteins are bioactive.
L58. Food proteins and not foodborne proteins
L61-62. It is not because prolamins can be hydrolyzed to release peptides with activities that they are high quality bioactive source. How do you define a high-quality versus a low-quality source protein?
L65-70. Those amino acids are not hypoglycemic. They may be present in the sequence but that doesn’t make them hypoglycemic. The remaining of the paragraph lack clarity.
L72-73. Delete “and has the potential to be developed into a hypoglycemic 72 functional factor for health food”
L74. What is active peptide database?
L71-80. Unclear
Comments on the Quality of English LanguageMajor language change
Author Response
Dear Experts:
Thank you for your letter and for the reviewers’ comments concerning our manuscript entitled “Screening and activity analysis of α-glucosidase inhibitory peptides derived from coix seed prolamins using bioinformatics and molecular docking” (ID: foods-2641995). Those comments are all valuable and very helpful for revising and improving our paper, as well as the important guiding significance to our researches. We have studied comments carefully and have made correction which we hope meet with approval.
Improved portion are marked in red in the revised manuscripts. Because the most current version has added some words, this caused the line numbers has been changed. The main corrections in the paper and the responds to the editor’s and review’s comments are in following next page. Expect the editor to allow the request.
If you require any additional information regarding our manuscript, please do not hesitate to contact us directly via the resources below. Thank you for your time and consideration.
Thank you and best regards.
Sincerely,
Zhi-Ming Li
Heilongjiang Bayi Agricultural University, 163319, Daqing, China
E-mail: lizhiming1998@126.com
Responds to the reviewer’s comments:
Reviewer #1:
- Comment 1:There are many expressions in the introduction that are incorrect as authors are trying to create their own. Some of those inaccurate expressions are provided in my assessment but the list is not exhaustive. Revision and rewriting are required
Answer: Thank you for your comments. We have carefully revised the areas that you pointed out as needing revision and have examined the "Introduction" section to correct any inappropriate content or expressions. The changes have been highlighted in red, please refer to the "Introduction" section.
- Comment 2:The structure of sentences in general is poor as the same work or expression is often repeated within the same sentence or the next.
Answer: Thank you for your suggestion. For this issue, the manuscript was scrutinized and sentences with repetitive expressions were streamlined or deleted. Please refer to the places marked in red in the manuscript.
- Comment 3:The rationale of testing the effects of sugars and mineral is not included in the manuscript. May not some information of that in the introduction.
Answer: Thank you for your suggestion. We have added this section to the "introduction" of the manuscript. It is emphasized that hydrolysates will encounter various food processing conditions when applied to food, and it is necessary to investigate the activity stability of hydrolysates. Please refer to lines 73 to 77 of the manuscript.
- Comment 4:How the previous work on the inhibition of α-glucosidase activity by hydrolysate of coix seed prolamins different from work included in the manuscript?
Answer: Thank you for your suggestion. In our previous work, we mainly focused on the preparation of the high α-glucosidase inhibitory activity of hydrolysates of coix seed prolamins (CHPs). The coix seed prolamins was treated by heat treatment combined with ultrasonic technology, and the CHPs was obtained by hydrolysis. The inhibitory activity, molecular weight distribution and in vitro digestion characteristics of CHPs were studied (Li, Z. M.; Zhang, S.; Bai, L.; Tang, H. C.; Zhang, G. F.; Zhang, J. Y.; Meng, W. M.; Zhang, D. J. Flexible processing technology of coix seed prolamins by combined heat-ultrasound: Effects on their enzymatic hydrolysis char-acteristics and the hypoglycemic activities of derived peptides. Ultrason. Sonochem. 2023, 98, 106526.).
In this study, mass spectrometry and bioinformatics techniques were used to further identify the composition of peptides that exert α-glucosidase inhibitory activity in CHPs, and molecular docking techniques were used to initially explore the mechanism of action. Finally, in order to broaden the application of CHPs in food, the activity stability of CHPS under common food processing conditions was also investigated in this study. To sum, this study is an extension of previous research.
- Comment 5:There of the hydrolysates had equivalent a-amylase inhibitory activities, only one was selected for further investigation without explanation.
Answer: Thank you for your suggestion. We guess what you mean is that the analysis in "3.1 The DH and α-glucosidase inhibitory activity of CHPs generated by different proteases" did not specify why the hydrolysate of coix seed prolamins prepared by alcalase protease was selected for the next test. First of all, proteases have enzymolysis specificity, and the peptide composition of the product of the same protein after hydrolysis by different proteases is different, even if the hydrolysate has the same α-glucosidase inhibitory activity. It is found that the specific enzymatic hydrolysis of some amino acids can help to obtain specific sequence of peptides, so as to achieve the improvement of activity. In our study, the degree of hydrolysis of CHPs and the inhibitory activity of α-glucosidase obtained by alcalase protease hydrolysis were both at the maximum. Probably because alcalase is an endoprotease with broad specificity, preferentially cleaving C-terminal hydrophobic amino acid residues such as Leu, Ile, Typ, Val, Phe, and Met. In addition, the content of "Leu + Ile + Val + Phe + Met" in coix seed prolamins accounted for 31.93% . This amino acid composition is conducive to the efficient hydrolysis of alcalase, and the polypeptide obtained via cleavage has a higher probability of conforming to the structural characteristics of α-glucosidase inhibitory peptides, resulting in high hydrolysis efficiency and high activity. In addition, we will consider further studies of other hydrolysates in the future.
- Comment 6:pH stability: The hydrolyzed proteins were adjusted to 2 – 12 and then kept at room temperature (25 ℃) for 30 min and brought back to pH 7.0. Because they are assays at the same pH, one should not typically expect a difference. Lowering or increasing a pH and then bringing it back to the initial value should not typically affect the structures of peptides/proteins.
Answer: Thank you for your question. This part is to first adjust the test environment to the corresponding pH value, and return to pH7.0 after processing. In fact, the same method has been used in other studies to investigate the structural and functional properties of proteins caused by pH shift. Xu et al. studied the effects of pH changes on physicochemical and structural properties of silkworm pupa protein isolates, It is found that alkaline pH shift combined with heat treatment altered the micromorphology of SPPI and destroyed the disulfide bonds between macromolecular subunits (72 and 95 kDa), resulting in reduced particle size and increased zeta potential and free sulfhydryl content of the isolates. Yang et al. found that the fluctuation of pH had an impact on the α-glucosidase inhibitory activity of Hot-Pressed Peanut Meal Protein Hydrolysates. During pH shift treatment, proteins will partially unfold under extreme alkaline or acidic conditions treating, and can refold when pH is adjusted back to neutrality. And the refolding does not restore its initial conformation, but creates a new “molten globule”, which makes the structure and functional traits of proteins vary ( https://doi.org/10.1016/j.foodchem.2020.126921).The references and relevant test data are as follows.
Xu H, Pan J, Dabbour M, et al. Synergistic effects of pH shift and heat treatment on solubility, physicochemical and structural properties, and lysinoalanine formation in silkworm pupa protein isolates[J]. Food Research International, 2023, 165: 112554. https://doi:10.1016/j.foodres.2023.112554 |
|
Protein solubility:
|
Thermal stability:
|
Average particle size:
|
FTIR spectra:
|
Zhang J, Liu Q, Chen Q, et al. Synergistic modification of pea protein structure using high-intensity ultrasound and pH-shifting technology to improve solubility and emulsification[J]. Ultrasonics Sonochemistry, 2022, 88: 106099. https://doi:10.1016/j.ultsonch.2022.106099 |
Intrinsic emission fluorescence spectra:
|
Sulfhydryl groups (-SH) and disulfide bond (S–S):
|
Fourier transform infrared spectroscopy:
|
Yang X, Wang D, Dai Y, et al. Identification and Molecular Binding Mechanism of Novel α-Glucosidase Inhibitory Peptides from Hot-Pressed Peanut Meal Protein Hydrolysates[J]. Foods, 2023, 12(3): 663. https://doi:10.3390/foods12030663 |
α-glucosidase inhibitory stability of hot-pressed peanut meal protein hydrolysates (PMHs) against pH:
|
- Comment 7:L34-36. Sentences should be reviewed to avoid repetition. E.g. the number of patients is rising … it is estimated that the number of patients or main type of diabetes … incidence of diabetes.
Answer: Thanks to your comments, We have adjusted the repetition here to make the whole statement more concise. Please refer to line 33 of the manuscript. At the same time, we examined the manuscript and made changes where there were similar problems.
- Comment 8: The meaning is unclear: Exogenous regulators to regulate and intervene.
Answer: Thank you for your advice. The term "Exogenous regulators" in this sentence is indeed an inappropriate expression. We have replaced it with the term "food-derived active ingredient" to make it more appropriate to the topic of this article and to highlight the role of food-derived components in the improvement of T2DM. For details see line 36 in the manuscript.
- Comment 9: I don’t understand the sentence: The deep excavation of their high-quality food sources is also key work in this field
Answer: Thank you for your comments. Here the author may not have made his intention clear. The main purpose here is to emphasize the importance of source to our previous investigations, highly active α-glucosidase inhibitory peptides have some similar structural features. These characteristics are mainly determined by the structure of the parent protein. Therefore, the acquisition of a good protein resource is the focus of the development of α-glucosidase inhibitory peptides. We have revised this sentence to make the meaning clearer and more coherent. Please refer to lines 48 to 50 of the manuscript for details.
- Comment 10: Delete coix seed before proteins and “the disease of”.
Answer: Thank you for your advice. We have deleted the word "the disease of" there. For details see line 55 in the manuscript.
- Comment 11: Intervening doesn’t seem to be the right expression.
Answer: Thank you for your comments. We have amended the "Intervening" here to "regulating" to make it more accurate. For details see line 55 in the manuscript.
- Comment 12:Delete “which has great research value”.
Answer: Thank you for your suggestion. We have removed this sentence. For details see line 56 in the manuscript.
- Comment 13:The main media expression is a wrong term to indicate dans peptides derived from proteins are bioactive.
Answer: Thank you for your suggestion. Our statement here is not accurate enough. What we want to express is that food proteins are hydrolyzed into peptides by various metabolic enzymes after ingestion. It then plays various physiological roles in the body in the form of peptides. In response, we reworked and revised the sentence to more accurately reflect what we were trying to say. Please refer to lines 56 to 58 of the manuscript for details.
- Comment 14: Food proteins and not foodborne proteins
Answer: Thank you for your suggestion. We have corrected the error there. For details see line 59 in the manuscript.
- Comment 15:L61-62. It is not because prolamins can be hydrolyzed to release peptides with activities that they are high quality bioactive source. How do you define a high-quality versus a low-quality source protein?
Answer: Thank you for your suggestion. We accept your suggestion that the previous describe is not accurate enough. The present study shows that the coix seed prolamins is a bioactive peptide source. The prepared peptides have the physiological functions of lowering blood pressure, protecting liver, lowering blood sugar, protecting vascular endothelial cells and so on. Coix seed prolamins has great prospect in developing functional active peptides, and is a source of multifunctional biological peptides. For this reason, we have revised this sentence to make it more accurate to express the author's intention. Please refer to lines 61 to 63 of the manuscript.
- Comment 16:L65-70. Those amino acids are not hypoglycemic. They may be present in the sequence but that doesn’t make them hypoglycemic. The remaining of the paragraph lack clarity
Answer: Thank you for your suggestion. This is a case of inaccuracy. We have revised this sentence. Ultimately, emphasizing the presence of these amino acids in bioactive peptides will contribute to the enhancement of hypoglycemic activity, as shown in lines 65-68 of the manuscript.
- Comment 17:L72-73. Delete “and has the potential to be developed into a hypoglycemic 72 functional factor for health food”
Answer: Thank you for your suggestion. We have deleted this sentence.
- Comment 18: What is active peptide database?
Answer: The database referred to here by the authors refers to the software and database for the sequence analysis of peptides after the hydrolysis is separated by mass spectrometry. Here are MaxQuant (version 2.1.2.0) software and UniProt database. We have made adjustments to this section to make it clearer and easier for readers to read. This database is described in detail in "2.6 Identification of the peptides (LC-MS/MS)"
- Comment 19:L71-80. Unclear
Answer: Thank you for your suggestion. We have modified this part of the content. According to the design idea of this study, the expression order is reorganized to make this part more clear, accurate and easy for readers to read. The specific has been marked in red in the manuscript. Please refer to lines 78 to 84 of the manuscript.
- Comment 19:Comments on the Quality of English Language: Major language change
Answer: Thank you for your suggestion. We have carried out professional English polishing on MDPI platform according to your suggestion. The supporting materials are shown in the figure below.
|

Reviewer 2 Report
Comments and Suggestions for Authors
Dear Authors,
Thank you for your valuable scientific contribution in the manuscript ID: foods-2641995
I have some suggestions, as follows:
- Line 27: “hypoglycemic peptide derived from coix seed prolamins”, I suggest “hypoglycemic peptide; coix seed prolamins” (I marked with yellow)
- Line 63: “ACE inhibitory”, I suggest to mention “Angiotensin-converting-enzyme inhibitors” (I marked with yellow)
- Line 71: “coix seed prolamins”, because it is for the first time in text I suggest to introduce the abbreviation “coix seed prolamins (CHPs)” (I marked with yellow)
- Line 94: “NaAc”, I suggest the substance name details (I marked with yellow)
- Table 1 and Line 145: “PBS” or “BPs”, please details of the abbreviation (I marked with yellow)
- Table 1: “NaCO3” , I suggest “Na2CO3” (I marked with yellow)
- Lines 145, 168, 170, 171, 173, 178, 196, 226, 228, 234, 251, 252, 254, 296, 332, 335: “CPHs” or “CHPs”, Please, uniformization the abbreviation of the term “coix seed prolamins” to be more clear, (I marked with yellow)
- Please, in the entire manuscript text, I suggest you to insert a blank between the number and the unit of measure (I marked with yellow in the specific lines)
- Please, in the entire text of the manuscript, I suggest you to insert a blank between the pH and its value (I marked with yellow in all lines)
- At References (Lines 466, 490): I recommend to use Italic Font for the scientific names of the species, in Latin language (I marked with yellow)
- At Reference [22], is missing the title of the article (I marked with yellow)
- Line 544: “Foods”, I suggest Italic Font “Foods” (I marked with yellow)
- Lines 559-560: “Food Res. Int.”, I suggest Italic Font “Food Res. Int.” (I marked with yellow)
- At References, the years mentioned of the articles, must be write with Bold Font, as the MDPI recommendations.
Thank you!
Best regards!

Minor editing of English language required.
Author Response
Dear Experts:
Thank you for your letter and for the reviewers’ comments concerning our manuscript entitled “Screening and activity analysis of α-glucosidase inhibitory peptides derived from coix seed prolamins using bioinformatics and molecular docking” (ID: foods-2641995). Those comments are all valuable and very helpful for revising and improving our paper, as well as the important guiding significance to our researches. We have studied comments carefully and have made correction which we hope meet with approval.
Improved portion are marked in red in the revised manuscripts. Because the most current version has added some words, this caused the line numbers has been changed. The main corrections in the paper and the responds to the editor’s and review’s comments are in following next page. Expect the editor to allow the request.
If you require any additional information regarding our manuscript, please do not hesitate to contact us directly via the resources below. Thank you for your time and consideration.
Thank you and best regards.
Sincerely,
Zhi-Ming Li
Heilongjiang Bayi Agricultural University, 163319, Daqing, China
E-mail: lizhiming1998@126.com
Responds to the reviewer’s comments:
Reviewer #1:
- Comment 1:Line 27: “hypoglycemic peptide derived from coix seed prolamins”, I suggest “hypoglycemic peptide; coix seed prolamins” (I marked with yellow)
Answer: Thank you for your comments. We have made the changes you suggested. And marked in red in the manuscript. Please refer to line 27 of the manuscript.
- Comment 2:Line 63: “ACE inhibitory”, I suggest to mention “Angiotensin-converting-enzyme inhibitors” (I marked with yellow)
Answer: Thank you for your suggestion. We have made the changes you suggested. And marked in red in the manuscript. Please refer to lines 64 to 65 of the manuscript.
- Comment 3:Line 71: “coix seed prolamins”, because it is for the first time in text I suggest to introduce the abbreviation “coix seed prolamins (CHPs)” (I marked with yellow)
Answer: We are very sorry and thank you very much for catching our mistake. We have made the changes you suggested. Please refer to lines 76 to 77 of the manuscript.
- Comment 4:Line 94: “NaAc”, I suggest the substance name details (I marked with yellow)
Answer: Thank you for your suggestion. The term "NaAc" here refers specifically to Sodium acetate, which we have already added here in the manuscript. Please refer to line 100 of the manuscript.
- Comment 5:Table 1 and Line 145: “PBS” or “BPs”, please details of the abbreviation (I marked with yellow)
Answer: Thank you for your suggestion. We have supplemented the specific details of PBS and BPs, as shown in Table 1 and line 151 of the manuscript..
- Comment 6:Table 1: “NaCO3” , I suggest “Na2CO3” (I marked with yellow)
Answer: We are very sorry and thank you very much for catching our mistake. We have made the changes you suggested. And marked in red in the manuscript. For details see Table 1 in the manuscript.
- Comment 7:Lines 145, 168, 170, 171, 173, 178, 196, 226, 228, 234, 251, 252, 254, 296, 332, 335: “CPHs” or “CHPs”, Please, uniformization the abbreviation of the term “coix seed prolamins” to be more clear, (I marked with yellow)
Answer: Thanks to your comments, The abbreviation we use for the Coix seed protein hydrolysate is CHPs. Thank you again for pointing out this issue, and we have corrected the errors you pointed out. Details of the changes have been marked in red in the manuscript.
- Comment 8:Please, in the entire manuscript text, I suggest you to insert a blank between the number and the unit of measure (I marked with yellow in the specific lines)
Answer: Thank you for your advice. We have corrected this problem by checking the full text. Please refer to the red marks in the manuscript for specific modification details. See lines 100, 110, 111, 112, 178, 190, 191, 386, and 388 of the manuscript for details.
- Comment 9:Please, in the entire text of the manuscript, I suggest you to insert a blank between the pH and its value (I marked with yellow in all lines)
Answer: Thank you for your comments. We have corrected this problem by checking the full text. Please refer to the red marks in the manuscript for specific modification details. See lines 114, 115, 157, 270, 361, 364 and 443 of the manuscript for details.
- Comment 10:At References (Lines 466, 490): I recommend to use Italic Font for the scientific names of the species, in Latin language (I marked with yellow)
Answer: Thank you for your advice. We have made the changes. Please look at lines 489 to 490 and 513 of the manuscript.
- Comment 11:At Reference [22], is missing the title of the article (I marked with yellow)
Answer: Thank you for your comments. We have added the corresponding title to this reference. Please refer to lines 527 to 529 of the manuscript.
- Comment 12:Line 544: “Foods”, I suggest Italic Font “Foods” (I marked with yellow)
Answer: Thank you for your suggestion. We have made the changes you suggested. Please refer to line 568 of the manuscript.
- Comment 13:Lines 559-560: “Food Res. Int.”, I suggest Italic Font “Food Res. Int.” (I marked with yellow)
Answer: Thank you for your suggestion. We have made the changes you suggested. Please refer to lines 589 to 590 of the manuscript
- Comment 14:At References, the years mentioned of the articles, must be write with Bold Font, as the MDPI recommendations.
Answer: Thank you for your suggestion. We have capitalized the year of the reference in the reference. Changes are marked in red, please refer to the "References" section.
- Comment 15:Minor editing of English language required
Answer: Thank you for your suggestion. We have carried out professional English polishing on MDPI platform according to your suggestion. The supporting materials are shown in the figure below.
|

Reviewer 3 Report
Comments and Suggestions for Authors
This manuscript represents an interesting study on the production and subsequent characterization of peptides capable of inhibiting glucosidase, and therefore, with potential anti-diabetic activity. The experimental approach is appropriate, the results are presented clearly, and the conclusions appear to be consistent with them.
Just some minor typographical indications:
Line 90: I'm sorry, but I couldn't find neutral protease or compound protease enzymes in the Sigma Aldrich catalog. Could you please provide more details about these enzymes?
Line 100: Do you not think it would be more appropriate to use 'precipitate' instead of 'precipitation'?
Line 244: Please change “α-glucosidas” to “α-glucosidase”.
Line 299: Change “energiies” to “energies”
Author Response
Dear Experts:
Thank you for your letter and for the reviewers’ comments concerning our manuscript entitled “Screening and activity analysis of α-glucosidase inhibitory peptides derived from coix seed prolamins using bioinformatics and molecular docking” (ID: foods-2641995). Those comments are all valuable and very helpful for revising and improving our paper, as well as the important guiding significance to our researches. We have studied comments carefully and have made correction which we hope meet with approval.
Improved portion are marked in red in the revised manuscripts. Because the most current version has added some words, this caused the line numbers has been changed. The main corrections in the paper and the responds to the editor’s and review’s comments are in following next page. Expect the editor to allow the request.
If you require any additional information regarding our manuscript, please do not hesitate to contact us directly via the resources below. Thank you for your time and consideration.
Thank you and best regards.
Sincerely,
Zhi-Ming Li
Heilongjiang Bayi Agricultural University, 163319, Daqing, China
E-mail: lizhiming1998@126.com
Responds to the reviewer’s comments:
Reviewer #3:
- Comment 1:Line 90: I'm sorry, but I couldn't find neutral protease or compound protease enzymes in the Sigma Aldrich catalog. Could you please provide more details about these enzymes?
Answer: Thank you for your comments. This belongs to the author's writing error, and the purchase manufacturer of protease is written wrong. In fact, all proteases were purchased from the Beijing Solarbio Technology Co., Ltd. (Beijing, China). We have modified the manufacturer information. In addition, we added detailed information of various proteases, such as CAS number and enzyme activity. Please refer to lines 93 through 97 of the manuscript. The neutral protease (CAS: 9068-59-1, 200 U/mg) and compound protease (CAS: 9014-01-1, 100 U/mg).
- Comment 2:Line 100: Do you not think it would be more appropriate to use 'precipitate' instead of 'precipitation'?
Answer: Thank you for your suggestion. This does make it clearer and more straightforward. We have made the changes you suggested. And marked in red in the manuscript, as shown in line 106 of the manuscript.
- Comment 3:Line 244: Please change “α-glucosidas” to “α-glucosidase”.
Answer: Thank you for your suggestion. We have made the changes you suggested. And marked in red in the manuscript, as shown in line 251 of the manuscript.
- Comment 4:Line 299: Change “energiies” to “energies”
Answer: Thank you for your suggestion. We have made the changes you suggested. And marked in red in the manuscript, as shown in line 306 of the manuscript..

Reviewer 4 Report
Comments and Suggestions for Authors
The topic was interesting, and it was well-designed. Each section was evaluated in terms of journal rules. The references were suitable for the topic. But the points should be revised as below;
Graphical abstract can be added.
The conclusion section can be improved by future expectations.
pH effects should be discussed deeply.
Edible sugar, NaCl, and metal ions sections in the results part were discussed with just 2 references for each. It should be enlarged.
Author Response
Dear Experts:
Thank you for your letter and for the reviewers’ comments concerning our manuscript entitled “Screening and activity analysis of α-glucosidase inhibitory peptides derived from coix seed prolamins using bioinformatics and molecular docking” (ID: foods-2641995). Those comments are all valuable and very helpful for revising and improving our paper, as well as the important guiding significance to our researches. We have studied comments carefully and have made correction which we hope meet with approval.
Improved portion are marked in red in the revised manuscripts. Because the most current version has added some words, this caused the line numbers has been changed. The main corrections in the paper and the responds to the editor’s and review’s comments are in following next page. Expect the editor to allow the request.
If you require any additional information regarding our manuscript, please do not hesitate to contact us directly via the resources below. Thank you for your time and consideration.
Thank you and best regards.
Sincerely,
Zhi-Ming Li
Heilongjiang Bayi Agricultural University, 163319, Daqing, China
E-mail: lizhiming1998@126.com
Responds to the reviewer’s comments:
Reviewer #4:
- Comment 1:Graphical abstract can be added.
Answer: Thank you for your comments. We have created the graphical abstract and uploaded the graphical abstract according to the file upload requirements. The graphical abstract is shown below:
|
- Comment 2:The conclusion section can be improved by future expectations.
Answer: Thank you for your suggestion. In the conclusion part, we prospected that the hypoglycemic effect of CHPs in vivo and its action pathway could be further analyzed through animal experiments in the future. Please refer to lines 449 to 452 of the manuscript for details of the revision.
- Comment 3:pH effects should be discussed deeply.
Answer: Thank you for your suggestion. In this section, we further analyze that the decrease of α-glucosidase inhibition activity in hyperacidic or hyperalkaline environments may be due to the disruption of the amino acid side chain of polypeptides. Please refer to lines 365 to 369 of the manuscript for details.
- Comment 4:Edible sugar, NaCl, and metal ions sections in the results part were discussed with just 2 references for each. It should be enlarged.
Answer: Thank you for your suggestion. We have extended in each section and further analyzed the reasons for the test results. The additions have been marked in red in the manuscript. Please refer to lines 397 to 399, 413 to 416, and 426 to 428 of the manuscript

Round 2
Reviewer 1 Report
Comments and Suggestions for Authors
There are still some issues with the science and the language. Some examples are provided below:
L34. “as the main type of diabetes”. Revise to: as the main type of the disease …
L35: “The typical characteristic of T2DM is persistent hyperglycemia or postprandial”. Revise to: Its typical characteristic is …
L36-37. “The food-derived active ingredients to regulate and intervene the blood glucose level in the body encompass an important strategy to improve T2DM”. Consider revising to: Finding food-derived active ingredients to regulate blood glucose level represent an important strategy to manage T2DM.
L40: “after the glucose participates in the systemic circulation”. Consider revising to: after the glucose is absorbed and enters the circulation system.
L48, L59. There are limited data to support the easy absorption of food derived peptides.
L209-212. It is incorrect to sate that the content of "Leu + Ile + Val + Phe + Met" is equivalent to the efficiency of alcalase hydrolysis. Those amino acids can be anywhere within the peptide sequences even there was little hydrolysis. The reference provided does not support that claim either. Authors are also stating the "Leu + Ile + Val + Phe + Met"
L211-214. Stating that the higher the "Leu + Ile + Val + Phe + Met" content, the higher the f α-glucosidase inhibition contradict authors own data as papain and trypsin hydrolysates with low DH had similar inhibitory activity as the alcalase hydrolysate.
L226. It is unclear what is the alkaline protease.
L233 – L…. The peptides were identified in one hydrolysate. The acronym CHPs is incorrect as it refers to all hydrolysates. Section tiles, text within the paragraphs, figure legends should be revised.
Comments on the Quality of English LanguageThe language needss revision
Author Response
Dear Experts:
Thank you for your letter and for the reviewers’ comments concerning our manuscript entitled “Screening and activity analysis of α-glucosidase inhibitory peptides derived from coix seed prolamins using bioinformatics and molecular docking” (ID: foods-2641995). Those comments are all valuable and very helpful for revising and improving our paper, as well as the important guiding significance to our researches. We have studied comments carefully and have made correction which we hope meet with approval.
Improved portion are marked in red in the revised manuscripts. Because the most current version has added some words, this caused the line numbers has been changed. The main corrections in the paper and the responds to the editor’s and review’s comments are in following next page. Expect the editor to allow the request.
If you require any additional information regarding our manuscript, please do not hesitate to contact us directly via the resources below. Thank you for your time and consideration.
Thank you and best regards.
Sincerely,
Zhi-Ming Li
Heilongjiang Bayi Agricultural University, 163319, Daqing, China
E-mail: lizhiming1998@126.com
Responds to the reviewer’s comments:
- Comment 1: “as the main type of diabetes”. Revise to: as the main type of the disease …
Answer: Thank you for your suggestion. We have revised this sentence according to your suggestion, which can be found in line 34 of the manuscript.
- Comment 2:L35:“The typical characteristic of T2DM is persistent hyperglycemia or postprandial”. Revise to: Its typical characteristic is …
Answer: Thank you for your suggestion. We have revised this sentence according to your suggestion, which can be found in line 35 of the manuscript.
- Comment 3:L36-37. “The food-derived active ingredients to regulate and intervene the blood glucose level in the body encompass an important strategy to improve T2DM”. Consider revising to: Finding food-derived active ingredients to regulate blood glucose level represent an important strategy to manage T2DM.
Answer: Thank you for your suggestion. We have revised this sentence according to your suggestion, which can be found in lines 36 to 37 of the manuscript.
- Comment 4:L40: “after the glucose participates in the systemic circulation”. Consider revising to: after the glucose is absorbed and enters the circulation system.
Answer: Thank you for your suggestion. We have revised this sentence according to your suggestion, which can be found in lines 40 to 41 of the manuscript.
- Comment 5:L48, L59. There are limited data to support the easy absorption of food derived peptides.
Answer: Thank you for your comments. We are not accurate enough with this statement. After reviewing some literature, we found that the bioavailability of bioactive peptides is actually related to the characteristics of the peptide. It can be depended on the peptide AA composition, number and type of AA residue, molecular size, structural features, net charge, pH, sequence, spatial conformation, the concentration of BAPs, condition of in vivo models, and molecular and surface hydrophobicity (Patil, P. J., Usman, M., Zhang, C., Mehmood, A., Zhou, M., Teng, C., & Li, X. (2022). An updated review on food-derived bioactive peptides: Focus on the regulatory requirements, safety, and bioavailability. Comprehensive reviews in food science and food safety, 21(2), 1732–1776. https://doi.org/10.1111/1541-4337.12911). Therefore, we have revised the two sentences you pointed out to ensure the accuracy of the narrative. Please refer to lines 46 to 48 and 57 to 60 of the manuscript.
- Comment 6:L209-212. It is incorrect to sate that the content of "Leu + Ile + Val + Phe + Met" is equivalent to the efficiency of alcalase hydrolysis. Those amino acids can be anywhere within the peptide sequences even there was little hydrolysis. The reference provided does not support that claim either. Authors are also stating the "Leu + Ile + Val + Phe + Met".
Answer: Thank you for your comments. Here, our analysis of the high hydrolysis efficiency of alcalase is not accurate enough. First, the efficiency of enzymolysis in enzymolysis tests depends largely on the affinity of proteins and proteases. The efficiency of enzymatic hydrolysis is related to the amino acid composition of protein, but it is not a decisive factor. The results of this study showed that the Alcalase possessed a stronger hydrolyzing ability to coix seed prolamins than other proteases. The reason was that Alcalase is an in-depth endoprotease and has more catalytic sites to break the peptide bonds in macroproteins, especially for hydrophobic and aromatic amino acids resides, such as Tyr, Trp, Phe, and Leu (Shazly, A. B., Mu, H., Liu, Z., El-Aziz, M. A., Zeng, M., Qin, F., Zhang, S., He, Z., & Chen, J. (2019). Release of antioxidant peptides from buffalo and bovine caseins: Influence of proteases on antioxidant capacities. Food chemistry, 274, 261–267. https://doi.org/10.1016/j.foodchem.2018.08.137). Therefore, we re-analyzed the reasons for the high efficiency of Acalase hydrolysis of Coix seed prolamins, which can be found in lines 208 to 211 of the manuscript.
- Comment 7:L211-214. Stating that the higher the "Leu + Ile + Val + Phe + Met" content, the higher the f α-glucosidase inhibition contradict authors own data as papain and trypsin hydrolysates with low DH had similar inhibitory activity as the alcalase hydrolysate.
Answer: Thanks to your comments. The analysis here is not accurate enough. Different proteases are specific, and the composition of the products obtained by hydrolyzing the coix seed prolamins is different. Because Alcalase has a wider range of enzyme cleavage sites, it may be the cause of higher enzymatic hydrolysis efficiency. In addition, protein hydrolysates with low degree of hydrolysis may also have high α-glucosidase inhibitory activity, which is closely related to the composition of peptides in hydrolysates. Some studies have shown a similar pattern. Yang et al. (Yang, X., Wang, D., Dai, Y., Zhao, L., Wang, W., & Ding, X. (2023). Identification and Molecular Binding Mechanism of Novel α-Glucosidase Inhibitory Peptides from Hot-Pressed Peanut Meal Protein Hydrolysates. Foods, 12(3), 663. https://doi.org/10.3390/foods12030663) found that although the degree of hydrolysis of hot-pressed peanut meal protein hydrolyzed by neutral protease was lower than that of Alcalase and Protamex, the inhibition rate of α-glucosidase was higher than that of the other two hydrolysates. Therefore, we have revised the content of this part, which can be found in lines 208 to 211 of the manuscript.
Yang, X., Wang, D., Dai, Y., Zhao, L., Wang, W., & Ding, X. (2023). Identification and Molecular Binding Mechanism of Novel α-Glucosidase Inhibitory Peptides from Hot-Pressed Peanut Meal Protein Hydrolysates. Foods . 12(3), 663. https://doi.org/10.3390/foods12030663 |
Comparison of degree of hydrolysis and α-glucosidase inhibition rate of different hydrolysates:
|
- Comment 8: It is unclear what is the alkaline protease.
Answer: Thank you for your advice. Thank you for your advice. We have made the changes, which can be found in line 224 of the manuscript.
- Comment 9:L233 – L…. The peptides were identified in one hydrolysate. The acronym CHPs is incorrect as it refers to all hydrolysates. Section tiles, text within the paragraphs, figure legends should be revised.
Answer: Thank you for your comments. Thank you for your advice. We have made the changes and the specific has been marked in red in the manuscript.
- Comment 10:Comments on the Quality of English Language: Moderate editing of English language required
Answer: Thank you for your suggestion. We have made some changes to the areas you pointed out that need English language improvement. In addition, we re-examine the language of the manuscript carefully to further ensure the accuracy of the English language expression.
